# SNAI2 cooperates with MEK1/2 and HDACs to suppress BIM- and BMF-dependent apoptosis in *TERT* promoter mutant cancers

Amol Tandon [1,2]*, Josh Lewis Stern [3,4,5]

**1** Department of Biochemistry and Molecular Genetics, University of Alabama at Birmingham, Birmingham, Alabama, United States of America, **2** Department of Medical and Translational Biology, Umeå University, Umeå, Sweden, **3** The Center for Clinical and Translational Science, University of Alabama at Birmingham, Birmingham, Alabama, United States of America, **4** Heersink School of Medicine, University of Alabama at Birmingham, Birmingham, Alabama, United States of America, **5** O'Neal Comprehensive Cancer Center, University of Alabama at Birmingham, Birmingham, Alabama, United States of America

* amol.tandon@umu.se

## Abstract

Cancers with *TERT* promoter mutations (TPM) display elevated RAS pathway signaling and mesenchymal traits, and associate with lower patient survival rates. We examined whether RAS pathway signaling in TPM cancers cooperates with mesenchymal features to drive resistance to apoptosis. We observed that RAS pathway signaling in TPM cancers inhibited apoptosis by downregulating the pro-apoptotic protein BIM. By using inhibitors of MEK1/2 kinases, we rescued the ability of TPM cancer cells to undergo apoptosis, which may have implications for targeted therapies. To further capitalize on this rescue, we explored combination treatments to drive apoptotic cell death. Treatment with the pan-BCL2 inhibitor, navitoclax (NX), in combination with MEK inhibition, significantly increased apoptosis, indicating that these cells are capable of undergoing intrinsic apoptosis, with BIM likely playing a critical role. Further, we found that transcriptional reprogramming of the mesenchymal state of TPM cancers using histone deacetylase inhibitors (HDACi) resulted in a synergistic increase in apoptosis, contingent upon BIM de-repression. Notably, the cause of this apoptosis appeared to be independent of DNA damage. The suppression of the mesenchymal transcription factor SNAI2, which has known roles in recruiting HDACs to silence gene expression, amplified apoptosis. Mechanistically, knockdown of SNAI2 impaired the cellular DNA repair leading to elevated basal levels of phosphorylated H2AX. Our findings show that TPM cancers exhibit specific small molecule vulnerabilities, driven by the convergence of RAS-MEK signaling and impaired HDAC regulation dependent on pro-apoptotic BH3-only proteins. Based on our findings, we propose that stratifying cancers based on TPM may identify a subset of tumors that are responsive to innovative combinations of inhibitors targeting these axes.

**Data availability statement:** All raw data generated in the study can be accessed in a Figshare repository with the following DOI: (10.6084/m9.figshare.28103435).

**Funding:** The author(s) received no specific funding for this work.

**Competing interests:** The authors have declared that no competing interests exist.

## Introduction

Genetic alterations in tumors play a crucial role in determining patient outcomes, and the development of personalized interventions is emerging as a promising therapeutic approach [1,2]. Coding mutations in genes such as *MYC*, *BRAF*, *TP53*, and *PIK3CA* are well known to drive a large subset of cancers [3–6].

Recent discoveries have highlighted the significance of non-coding mutations in cancer. For example, the promoter region of the *TERT* gene that encodes the catalytic subunit of telomerase is essential for telomerase expression in these cancers [7,8]. These *TERT* promoter mutations (TPM) are among the most common cancer driver mutations so far discovered [9]. TPM are particularly prevalent in glioblastomas, melanomas, myxoid liposarcomas, and cancers of the liver, thyroid, and bladder, and are often associated with poor patient survival [10–13]. TPM have been found to associate with a distinct gene expression profile characterized by elevated MAPK/RAS pathway signaling and a hybrid epithelial-to-mesenchymal (EMT) phenotype, which is observed in several different tissue types [14]. Both RAS signaling and a hybrid EMT signature are linked to therapy-resistance, metastasis, and tumor recurrence [15,16].

Histone deacetylase (HDAC) inhibitors have shown promise in reprogramming such EMT states in cancer, and epigenetic modifiers have demonstrated cooperativity with proapoptotic approaches and RAS pathway inhibition [17–19]. The EMT state in *TERT* promoter mutants is also driven by elevated levels of transcription factors such as ZEB1 and ZEB2, SNAI2, TWIST1 and TWIST2, among others [14]. In particular, SNAI2 (a.k.a. slug) is a MEK1/2-regulated transcription factor implicated in EMT, which is consistently elevated in TPM cancers. SNAI2 can play important roles in tumor aggressiveness, possibly by recruiting chromatin regulators for epigenetic silencing [20].

Cancer cells cause lethal disease in part due to defects in their programmed cell death pathways, such as intrinsic apoptosis. In this study, we investigated the mechanisms that prevent apoptotic cell death in TPM cancers. We found that pro-apoptotic BIM and BMF levels are suppressed by constitutive MEK1/2 signaling in these cancers and disrupting this pathway triggers BIM and/or BMF-dependent apoptosis across various TPM cancer types. Additionally, we showed that this apoptotic response is greatly enhanced by the use of a BH3 mimetic, which directly inhibits BCL-2 proteins, or by reprogramming gene expression with FDA-approved HDAC inhibitors. We also provide compelling evidence that the transcription factor SNAI2, in conjunction with MEK1/2 signaling, drives the hybrid EMT phenotype in TPM cancers, facilitating survival and apoptosis resistance through epigenetic suppression of apoptotic markers like BIM and BMF.

## Results

### MEK1/2 inhibition in TPM cancers promotes BIM rescue leading to apoptosis

Cancer cells frequently employ mechanisms that enable escape from apoptosis, including activation of RAS signaling [21,22]. *TERT* promoter mutant (TPM) cancer

cells are characterized by gene expression signatures matching RAS pathway driven states, although many of these are not explained by mutations known to activate the RAS pathway [14]. We first validated RAS pathway activation in a panel of TPM cancer cell lines by determining the level of phosphorylated ERK1 and ERK2 (pERK), aiming to document the frequent association of TPMs with elevated RAS signaling. This panel included melanoma (A101D, UACC257), hepatocellular carcinoma (SNU475, SNU423), breast cancer (MDA-MB-231), medulloblastoma (DAOY), and neuroblastoma (SKNSH), which represent diverse cancer types harboring TPMs. It should be noted that some of these lines (A101D, UACC257, MM231, and SKNSH) also carry activating mutations in the RAS/RAF/ERK pathway, as detailed in Table S1 in S1 File. Phosphorylated ERK was found to be strongly expressed in all the cell lines by immunoblot. Moreover, low dose of MEK1/2 inhibitor trametinib (TR; 25 nM) was able to inhibit the ERK activation in these cells (Supplementary Fig S1a in S1 File).

We also validated expression of mesenchymal markers such as vimentin, SNAI2, and ZEB1, which were found to be strongly expressed and accompanied by absence of epithelial markers like GRHL2 and E-cadherin in majority of the cell lines (Supplementary Fig S1b in S1 File). For comparison, we included the atypical TPM bladder cancer cell line SCaBER, which expressed SNAI2, but not other mesenchymal markers tested. Instead SCaBER expressed epithelial markers GRHL2, and E-cadherin (Supplementary Fig S1b in S1 File).

Upon treatment of cell lines from different cancer types with MEK1/2 inhibitor trametinib (MEKi, TR), BIM protein expression was rescued (Fig 1a). These included melanoma (Mel), hepatocellular carcinoma (HCC), breast cancer (BrCa), medulloblastoma (MB), neuroblastoma (NB), bladder cancer (BlCa) and glioblastoma (GBM). Consistent with these observations, analysis of reverse-phase-protein-array (RPPA) data of several hundred cell lines indicated that, with few exceptions, *TERT* promoter mutant cancer types maintain below average levels of endogenous BIM protein expression relative to cancers lacking mutant *TERT* promoter (Supplementary Fig S2a and b in S1 File).

We next assessed if MEKi-induced changes in BIM levels associated with increased apoptosis. We monitored cleaved caspase 3 (CC3) in cells treated with TR. For this purpose, we employed a melanoma line, A101D, with known mutations in the RAS pathway and an HCC line, SNU475, lacking any reported mutations in the RAS pathway (Table S1 in S1 File). BIM expression was elevated in both cell lines from 1 hour and increased up through 24 hours, and doses of 25 nM were sufficient to observe increased BIM (Fig 1b). These increases in BIM levels were matched by increasing levels of CC3 indicating that the treatment also triggered apoptosis. The increased apoptosis in these cell lines upon MEK inhibition is a direct consequence of increased BIM expression, which was confirmed by employing siRNA against BIM in TR-treated cell lines (U87, LN229, SNU475 and A101D). Upon treatment with BIM siRNA, we observed rescue the MEKi-mediated cleavage of caspase-3 in these cell lines (Fig 1c).

BIM is a pro-apoptotic member of the BCL-2 family that promotes cell death by binding to and inhibiting anti-apoptotic BCL-2 family proteins like BCL-2, BCL-XL, and MCL-1, thereby freeing pro-apoptotic effectors such as BAX and BAK to initiate mitochondrial outer membrane permeabilization and apoptosis [23,24]. Our results suggested that TPM cancers could be sensitized to intrinsic apoptosis. To more fully test this, we treated cells with the pan-BCL2 family inhibitor (ABT-263, navitoclax, NX) which inhibits three of the four major pro-survival BCL-2 family proteins (BCL-2, BCL-XL, and BCL-w) and promotes BAX/BAK dimerization and mitochondrial apoptosis [25]. We used NX alone or in combination with MEKi for 24 hours, followed by an assessment of apoptosis. NX modestly induced apoptosis on its own but when used in combination with MEKi nearly all cell lines displayed a pronounced increase in apoptosis (Fig 1d). Taken together, our data demonstrate that TPM cancer cells can be made competent to undergo intrinsic apoptosis when treated with MEKi and BCL-2 family inhibitors, and that BIM is likely a critical mediator of MEKi-induced apoptosis in cells.

## HDAC inhibitors cooperate with MEK1/2 inhibition to promote apoptosis in TPM cancer cell lines

Reprogramming gene expression of mesenchymal cancer cells using HDAC inhibitors has shown some promise in part by promoting apoptosis by upregulating BIM. Such BIM expression is key to inducing apoptosis in some mesenchymal lung

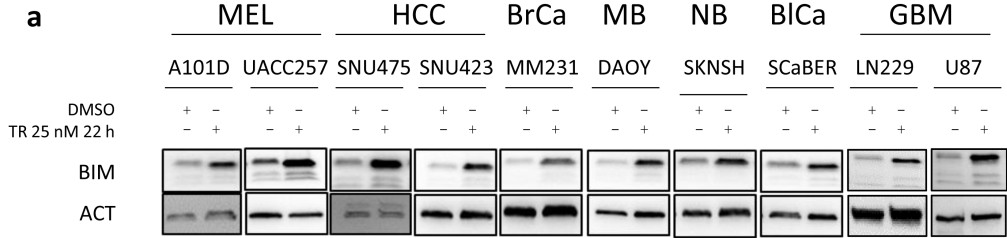

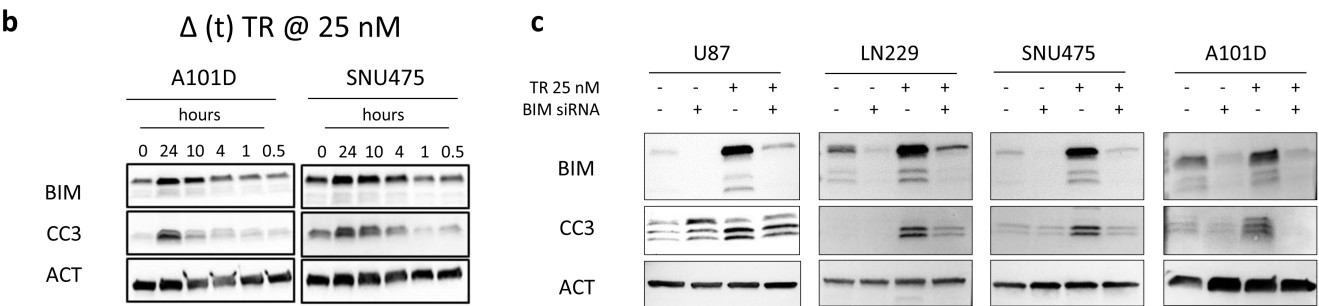

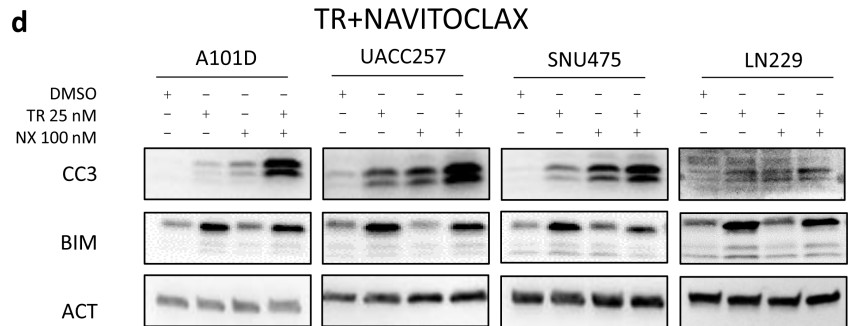

**Fig 1. MEK1/2 inhibition induces BIM-dependent apoptosis in *TERT* promoter mutant cancer cells.** (a) MEK1/2 was inhibited in cells by treatment with trametinib (TR) 25 nM for 22 hours followed by immunoblot for BIM isoforms. (b) MEK1/2 was inhibited in A10D (Melanoma) and SNU475 (Hepatocellular carcinoma) cells by treatment with TR 25 nM for varying amounts of time before harvest followed by analysis of the marker for apoptosis, cleaved caspase-3 (CC3) and BIM. (c) U87, LN229 (Glioblastoma), SNU475 (Hepatocellular carcinoma) and A101D (Melanoma) cells were treated with 50 pmol BIM siRNA (48 hours) ± TR (24 hours). Immunoblots show the levels of BIM and cleaved caspase-3. (d) A101D, UACC257 (both Melanoma) and SNU475 (Hepatocellular carcinoma) cells were treated with TR and pan-BCL2 inhibitor Navitoclax (NX) for 24 hours. Immunoblots indicate markers of apoptosis CC3, PARP and changes in BIM protein levels. ACT, β-Actin, used as loading control for immunoblots. MEL – Melanoma; HCC – Hepatocellular carcinoma; BrCa – Breast cancer; MB – Medulloblastoma; NB – Neuroblastoma; GBM – Glioblastoma.

and colon cancers [26,27]. Since TPM cancers display mesenchymal features, we sought to assess if HDAC inhibition (HDACi) could cooperate with MEKi to promote apoptosis. We tested a pan-HDAC inhibitor, vorinostat (VOR), which is FDA approved for use with cutaneous T-cell lymphoma [28]. VOR is effective at micromolar concentrations in various cell line models at inhibiting cell growth or inducing cell death. We tested 2.5 μM VOR either alone or combination with low dose TR in a panel of TPM cancer cells (melanoma (A101D, UACC257), hepatocellular carcinoma (SNU475, SNU423), breast cancer (MDA-MB-231), medulloblastoma (DAOY), neuroblastoma (SKNSH) and glioblastoma (LN229, U87)) followed by analysis of CC3 levels. In most cells tested, VOR on its own led to a moderate increase in CC3 (Fig 2a). In those lines, the addition of TR at 25 nM resulted in clear enhancement of apoptosis. In one of these lines, a medulloblastoma (DAOY) cell line, VOR on its own led to a marked increase in CC3, which was not enhanced by 25 nM TR.

**a**

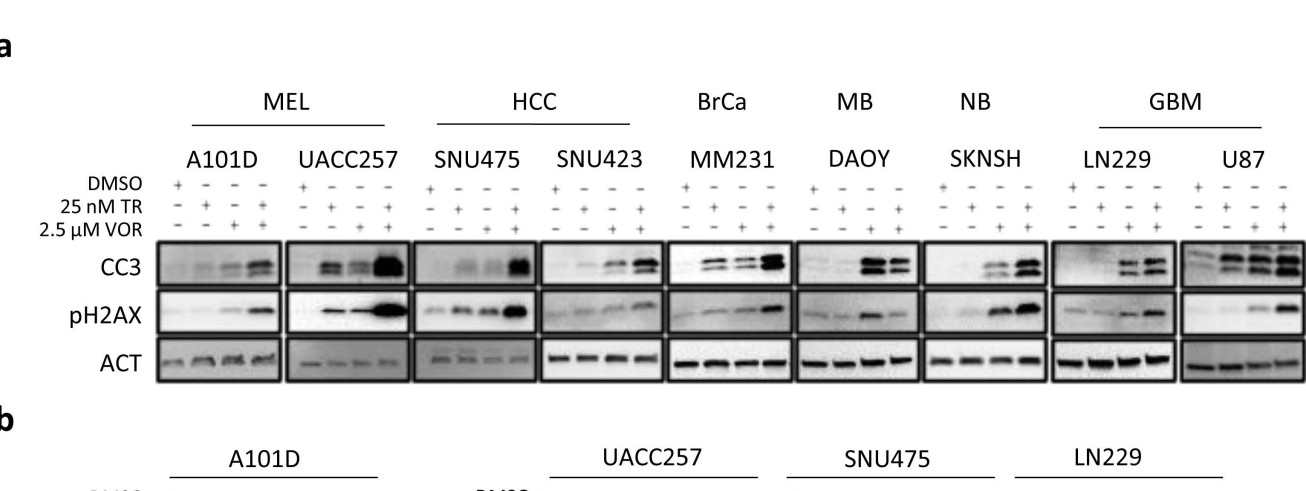

**b**

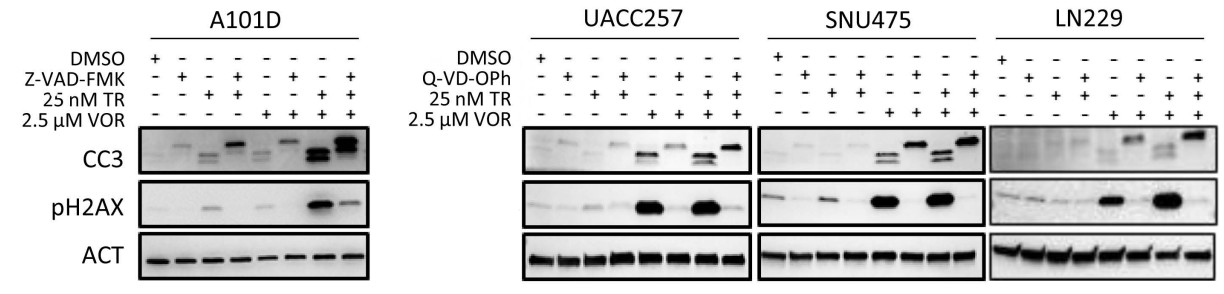

**c**

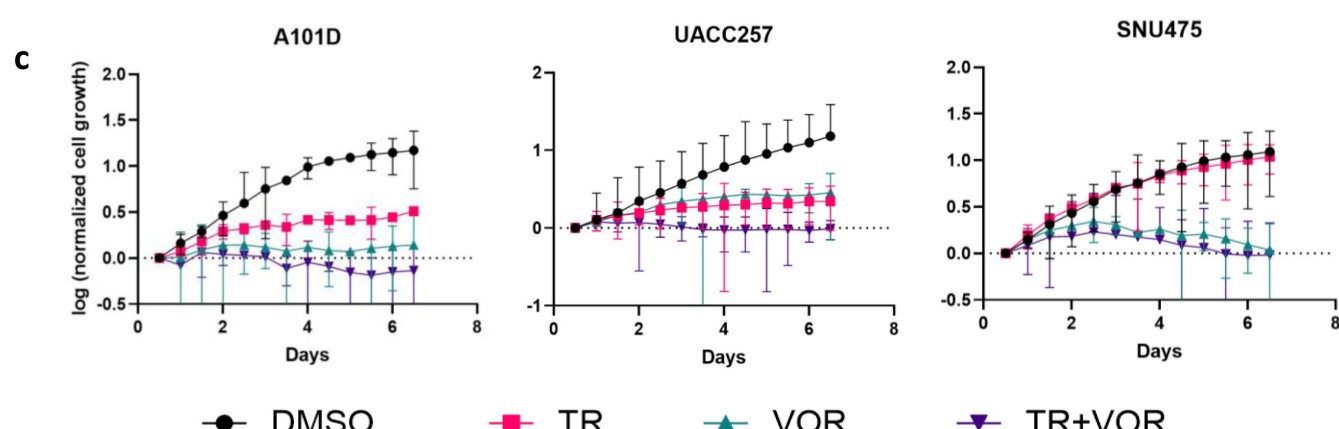

Fig 2. **MEK1/2 inhibition cooperates with HDAC inhibition to enhance apoptosis in multiple *TERT* promoter mutant cancer cells.** (a) Cells were treated for 24 hours with trametinib (TR) with or without vorinostat (VOR) and probed for apoptosis (CC3) and induction of DNA damage (pH2AX) by immunoblot. (b) A101D, UACC, SNU475 and LN229 cells were treated with inhibitors of MEK1/2 (TR 25 nM) and HDAC (VOR 2.5 µM) in presence of pan caspase inhibitors Z-VAD-FMK or Q-VD-OPh, 50 µM for 24 hours. Immunoblots show changes in markers of apoptosis (cleaved caspase 3 (CC3)) and DNA damage (pH2AX). (c) 5 x 103 cells were seeded in 96 well plates and after 12 hours treated with vehicle, or trametinib (TR), vorinostat (VOR) or both followed by automated cell counting using a BioSpa Cytation-5 for 6.5 days. β-Actin (ACT) was used as loading control for immunoblots. MEL – Melanoma; HCC – Hepatocellular carcinoma; BrCa – Breast cancer; MB – Medulloblastoma; NB – Neuroblastoma; GBM – Glioblastoma.

Histone deacetylase inhibitors induce hyperacetylation of the N-terminal tails of histones H3 and H4, leading to a more open chromatin structure. Such global epigenetic changes increase conflicts between replication and transcription machinery. This in turn leads to elevated DNA damage that triggers apoptosis [29]. This effect is compounded in tumor cells compared to most normal cells due to their relatively high rate of cell division. Consistent with HDAC inhibitor causing DNA damage, we observed upregulation of a DNA damage marker, phosphorylated Histone H2A-X (pH2AX) in cells

treated with VOR. Importantly, this effect of VOR on DNA damage was dramatically enhanced by treating with a combination with VOR + MEKi (Fig 2a).

These results suggested that these drugs caused DNA damage which triggered apoptosis. However, activation of caspases during apoptosis can also cause DNA fragmentation and pH2A-X accumulation. Therefore, to test that the observed DNA damage was not a result of caspase activation, we blocked caspase activation with two peptides (Z-VAD-FMK and Q-VD-Oph) in TR + VOR treated cells and examined levels of pH2AX. Surprisingly, in these caspase-inhibited controls, the pH2AX signal in TR + VOR treatments was almost completely blocked (Fig 2b). We interpret this observation to indicate that the DNA damage in Fig 2a is not primarily responsible for the apoptosis but was rather a consequence of it. This observation contrasts with previous studies which have indicated that H2AX phosphorylation is causal in cell death in melanoma cell lines treated with HDAC/MEK inhibitor combination [19].

As an alternative mechanism, we considered the possibility that adding VOR to TR treated cells may result in a greater rescue of BIM levels, leading to higher levels of apoptosis. Indeed, in several cell lines further increases in BIM were noted (Supplementary Fig S3a in S1 File). However, in a number of lines, no further increase was observed, suggesting alternative mechanisms.

In cells, the intracellular molar ratio of BIM to the pro-survival proteins BCL-2, BCL-XL, BCL-w and MCL-1 can be critical to the outcome of whether BIM is capable of triggering mitochondrial depolarization by activation of BAX/BAK heterodimers. Therefore, we assessed whether VOR decreased levels of pro-survival proteins, effectively increasing the molar ratio of BIM to the anti-apoptotic factors. In A101D, UACC257 and SNU475 (cell lines that did not previously display further increases in BIM upon VOR treatment) each displayed reductions in the levels of BCL-XL (Supplementary Fig S3b in S1 File). We conclude that these losses in pro-survival proteins are likely to potentiate BIM-driven apoptosis.

Our results suggest that this or a similar drug combination may be useful against these cancer types. However, it remained unclear whether these drugs impacted the growth of these tumor cells. To determine the combinatorial impact of TR+HDACi on growth of these cells, we seeded A101D, UACC257 and SNU475 cells in 96 well plates, and treated them with low doses of TR with or without HDAC inhibition (Fig 2c). Combinations of these two inhibitors markedly inhibited cell growth measured by automated cell counting (Fig 2c).

## Induction of apoptosis in TPM cancers by MEKi+HDACi is dependent on BIM and BMF

We previously reported that the inhibition of HDACs enhances MEK1/2 inhibitor (MEKi)-induced BIM and BMF dependent apoptosis in GBM [30]. We therefore sought to investigate whether similar mechanisms apply to other TPM cancers. To investigate the mechanism by which MEK1/2 inhibition cooperates with HDAC inhibition in TPM cancers, SNU475 (hepatocellular carcinoma) and LN229 (glioblastoma) cells were treated with clinically relevant MEK and HDAC inhibitors, including TR and VOR, along with other specific HDAC inhibitors: mocetinostat (MOC, HDAC 1/2/3 inhibitor, 1 μM), entinostat (ENT, HDAC 1/3 inhibitor, 2 μM), LMK235 (LMK, HDAC 4/5 inhibitor, 1 μM), and nexturastat (NEX, HDAC 6 inhibitor, 2 μM). Immunoblot analysis revealed that these treatments led to a clear increase in CC3 levels, as well as increased levels of BIM and/or BMF proteins in cells treated with MEKi+HDACi compared to either treatment alone (Fig 3a). These results were consistent with our previous findings in GBM, suggesting that HDAC inhibition potentiates MEKi-induced apoptosis across different types of TPM cancers and that this effect is driven by rescue of pro-apoptotic BIM and BMF protein levels.

To determine whether the upregulation of BIM and BMF is essential for the observed apoptotic response, we specifically silenced BIM and BMF expression in LN229, A101D (BIM and BMF knockdown) and SNU475 (BMF knockdown) cells. Following knockdown, cells were treated with TR (25 nM) and VOR (2.5 μM) for 24 hours. Knockdown of either BIM or BMF strongly attenuated CC3 levels compared to control siRNA-treated cells (Fig 3b,c), indicating that both proteins are critical for MEKi+HDACi-induced apoptosis.

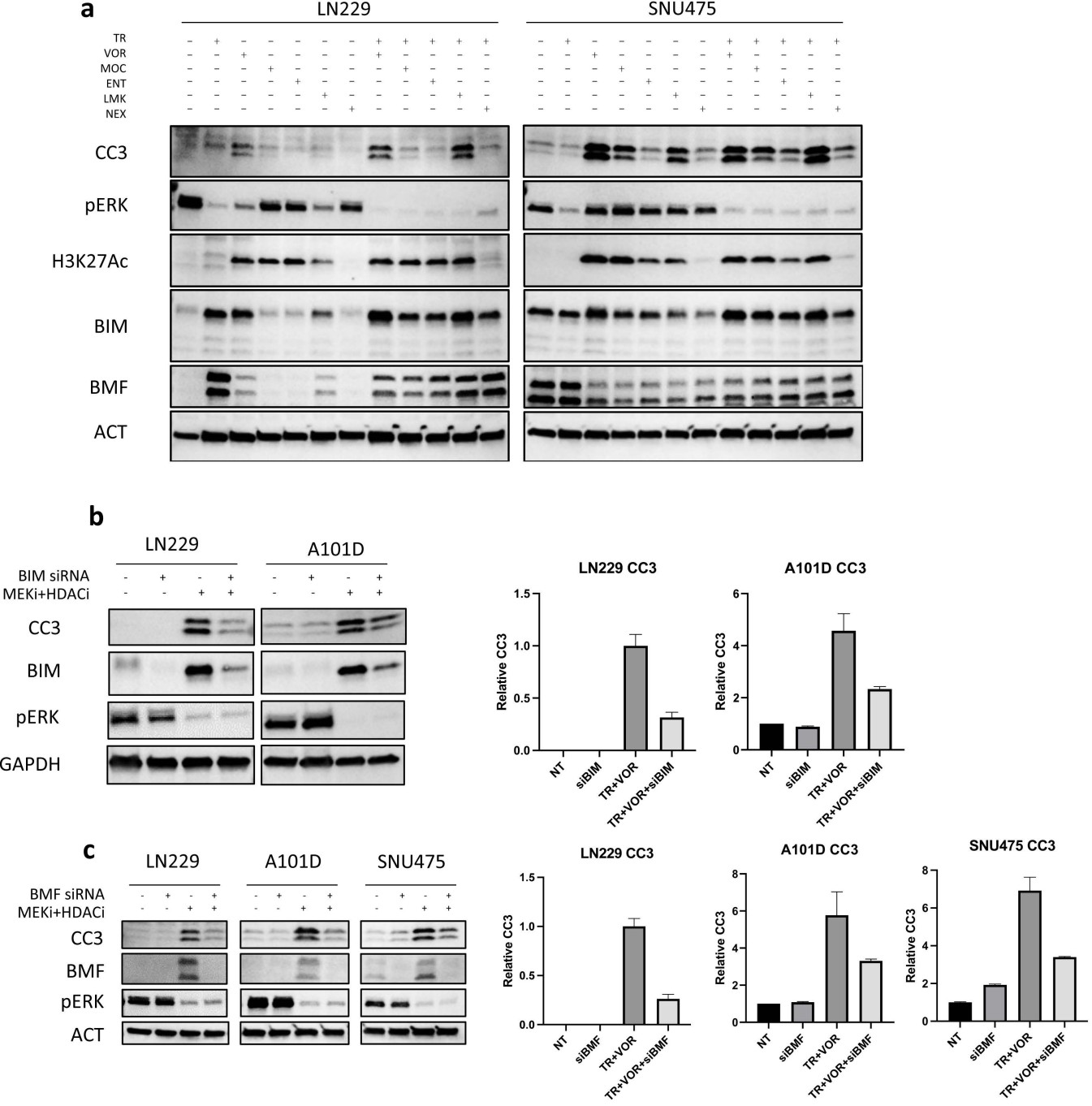

**Fig 3. Apoptosis induction by MEKi +HDACi is dependent on *BIM* and *BMF* de-repression.** (a) TPM cancer cells were treated with selective HDAC inhibitors. LN229 and SNU475 cells were treated with mocetinostat (MOC, a HDAC 1/2/3 inhibitor, 1 μM), entinostat (ENT, a HDAC 1/3 inhibitor, 2 μM), LMK235 (LMK, a HDAC 4/5 inhibitor, 1 μM) and nexturastat (NEX, a HDAC 6 inhibitor, 2 μM) for 24 hours with or without MEK inhibition (TR). Figure depicts immunoblots for markers of apoptosis (CC3, BIM and BMF). pERK and H3K27Ac were used to validate the effects of TR and HDAC inhibitors respectively. (b), (c) To test if the apoptosis induced by MEK1/2 inhibition + HDAC inhibition is dependent on BIM and BMF, cells were treated with 50 pmol BIM or BMF siRNA with or without 25 nM TR + 2.5 μM VOR followed by analysis of cleaved caspase 3 (CC3). Non-targeting siRNA pool was used as control. Also depicted in these panels is the quantification of CC3 (after normalization with actin) from three independent experiments. GAPDH or β-Actin were used as loading controls for immunoblots.

This highlights the role of BIM and BMF in mediating the apoptotic response induced by MEKi+HDACi in TPM cancers. The synergistic induction of apoptosis by this combination treatment is highly dependent on the de-repression and transcriptional activation of these pro-apoptotic genes.

### SNAI2 suppresses apoptosis by localizing to and promoting histone deacetylation of BIM and BMF proximal promoters

In addition to RAS pathway activation, TPM cancer cells display significantly elevated SNAI2 expression compared to wild-type (WT) cancers across a range of tumor types (Fig 4a). Given the well-documented association between elevated SNAI2 levels and mesenchymal phenotypes, this suggested a compelling link between SNAI2 expression and the aggressive phenotype observed in TPM cancers. To ascertain the role of SNAI2 in TPM cancers in promoting tumor cell survival, we used siRNA and overexpression approaches, coupled with measuring apoptosis. In all four TPM tumor lines tested, knockdown of SNAI2 using 50 pmol siRNA for 72 h led to a marked increase in apoptosis as measured by CC3 and cleaved PARP (CP) (Fig 4b). However, only A101 and SNU475 showed reductions in endogenous levels of apoptosis with overexpression of SNAI2 after addition of doxycycline.

SNAI2 can regulate cancer cell DNA damage responses [31], suggesting that SNAI2 may suppress apoptosis by enhancing DNA repair. To analyze the effect of SNAI2 suppression on apoptosis induction and DNA damage, we inhibited apoptosis using Z-VAD-FMK and analyzed pH2AX in A10D and U87 cells after SNAI2 knockdown. Blocking apoptosis reduced CC3 levels as well as pH2AX. We conclude that a strong driver of pH2AX elevation after loss of SNAI2 is due to apoptosis (Fig 4c). Importantly, however, the persistence of some pH2AX even after caspase inhibition suggests that SNAI2 suppression impairs DNA repair (Fig 4c).

As we showed that BIM is important for regulating cell death in TPM cancers, we considered that SNAI2 may also play a role in regulating BIM. We first examined BIM mRNA after SNAI2 siRNA treatment. SNAI2 knockdown resulted in a rescue of BIM mRNA levels across different cell lines, including LN229, A101D, and SNU475 (Fig 4d). Given the role that SNAI2 has in mediating epigenetic repression, these data suggest that SNAI2 may directly repress BIM transcription by binding to the BIM promoter. We tested this hypothesis using chromatin immunoprecipitation (ChIP) assays for SNAI2 at the BIM proximal promoter. We also examined the BMF proximal promoter for SNAI2 occupancy. These experiments indicated that SNAI2 localized to the promoters of both BIM and BMF, where it can potentially facilitate the recruitment of repressive histone modifying machinery such as polycomb repressive complex 2 (PRC2) and histone deacetylases (HDACs) (Fig 4e,f) to promote transcriptional repression. To test this, we examined cells treated with SNAI2 siRNA for loss of the repressive histone 3 lysine 27 trimethylation (H3K27me3, deposited by PRC2) and gain of H3K27 acetylation (catalyzed by histone acetyl transferases, associated with transcriptional activation). Cells treated with SNAI2 siRNA showed reduced recruitment of H3K27me3 to both BIM and BMF promoters. This reduction in repressive histone marks correlated with increased acetylation of histones, marked by H3K27Ac, indicating a more open chromatin configuration conducive to gene transcription. To determine the extent to which SNAI2 regulated global histone acetylation levels, we employed a siRNA-resistant-SNAI2-expressing plasmid. Transfection of U87 GBM cells, treated with SNAI2 siRNA together with this plasmid, resulted in a rescue of SNAI2 protein and an upregulation of global histone acetylation upon treatment with SNAI2 siRNA, which was nearly completely rescued by using the resistant plasmid (Fig 4g).

### SNAI2 knockdown enhances BIM/BMF dependent apoptosis in TR+VOR treated TPM cells

To further investigate the regulatory potential of SNAI2 in TPM cancer cell survival, we tested whether its expression limited the efficacy of MEKi and HDACi in inducing cell death. We tested the cooperative effects of SNAI2 depletion by combining it with TR+VOR followed by assessing apoptosis in TPM cells. Our observations suggested that the increased expression of BIM and BMF that results from SNAI2 knockdown may enhance apoptosis in TPM cancer cells treated with MEK1/2 and HDAC inhibitors (Fig 5a). The role of BIM and BMF in apoptosis induction upon SNAI2 siRNA+TR+VOR

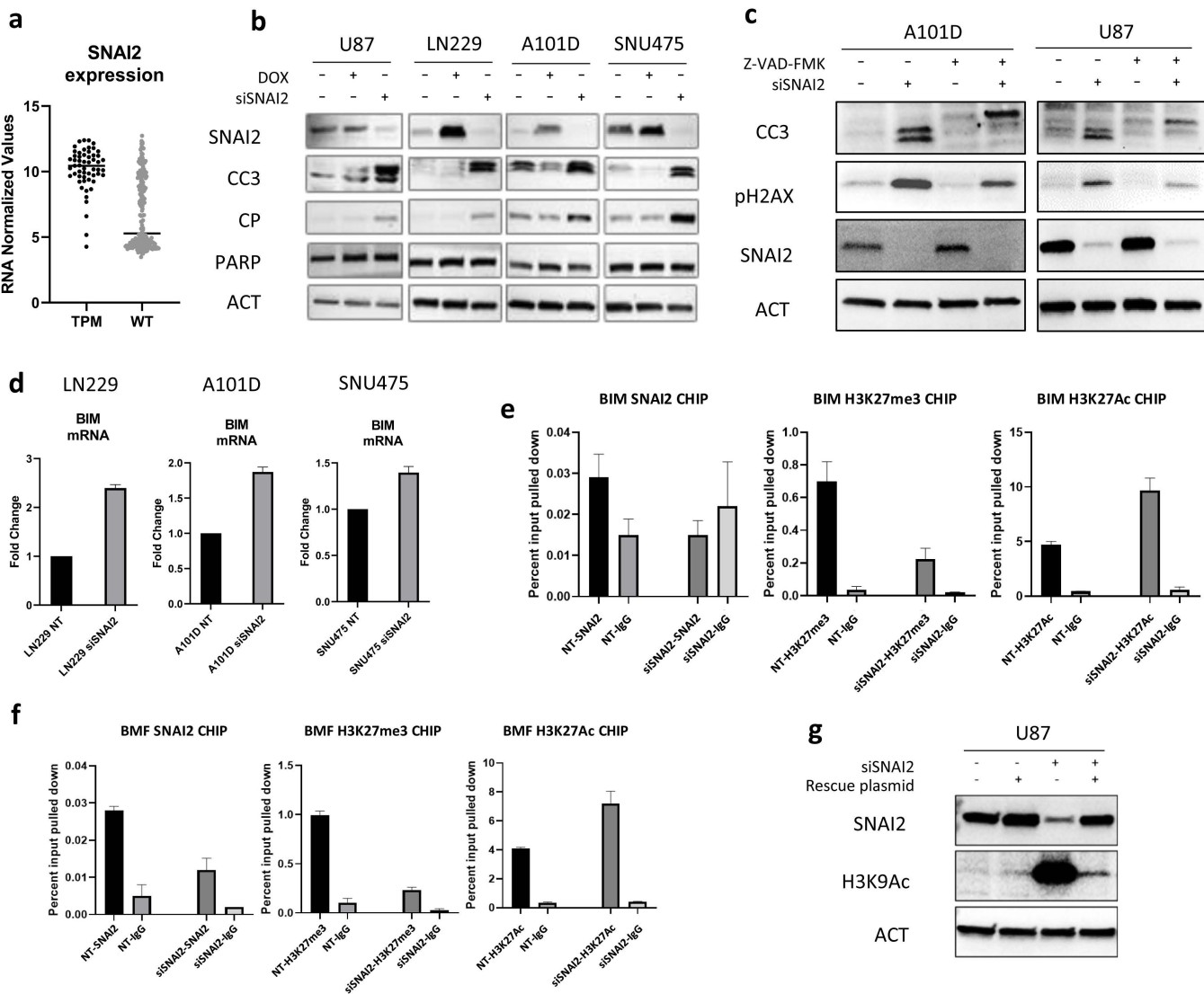

**Fig 4. SNAI2 suppresses apoptosis and regulates *BIM* and *BMF* proximal promoters.** (a) *SNAI2* RNA expression levels in *TERT* promoter mutants (TPM) cancer cell lines (n = 55) or wild type (WT) cancer cell lines (n = 207). Data are from Cancer Cell Line Encyclopedia (CCLE, Broad Institute). N = 55 TPM cell lines and 207 wild type cell lines. (b) Dox-inducible SNAI2 overexpressing cell lines were induced with 200 ng/ml doxycycline for 96 hours with or without 50 pmol siRNA targeting *SNAI2* (siSNAI2) to test the induction of apoptosis (cleaved caspase 3 (CC3) and cleaved parp (CP)) by immunoblots upon SNAI2 repression. (c) A101D and U87 cells were co-treated with the pan caspase inhibitors Z-VAD-FMK, 50 µM for the 96 hours, with or without siRNA against *SNAI2* and immunoblotted for apoptosis, DNA damage markers and SNAI2. (d) RT-PCR data showing rescue of *BIM* mRNA upon siRNA-based knockdown of *SNAI2* in LN229, A101D and SNU475 cells. (e,f) To test if SNAI2 localizes to *BIM* and *BMF* promoters and affects their expression, chromatin immunoprecipitation (ChIP) was performed. LN229 cells were treated with 50 pmol *SNAI2* siRNA (siSNAI2) or control non-targeting (NT) siRNA and cultured for four days. Sheared chromatin was precipitated with antibodies against SNAI2, H3K27me3, or H3K27Ac. Non-specific IgG was used as a negative control. Samples were analyzed by qRT-PCR using primers designed to amplify known/predicted SNAI2 binding sites in *BIM* and *BMF* promoter. ChIP samples were normalized to input chromatin. (g) U87 cells were treated with 50 pmol siRNA against *SNAI2* and rescued with a SNAI2-overexpressing plasmid that was resistant to the siRNA. Immunoblots depict rescue of SNAI2 and changes in histone-3 acetylation at K9 (H3K9ac). β-Actin was used as loading control for immunoblots.

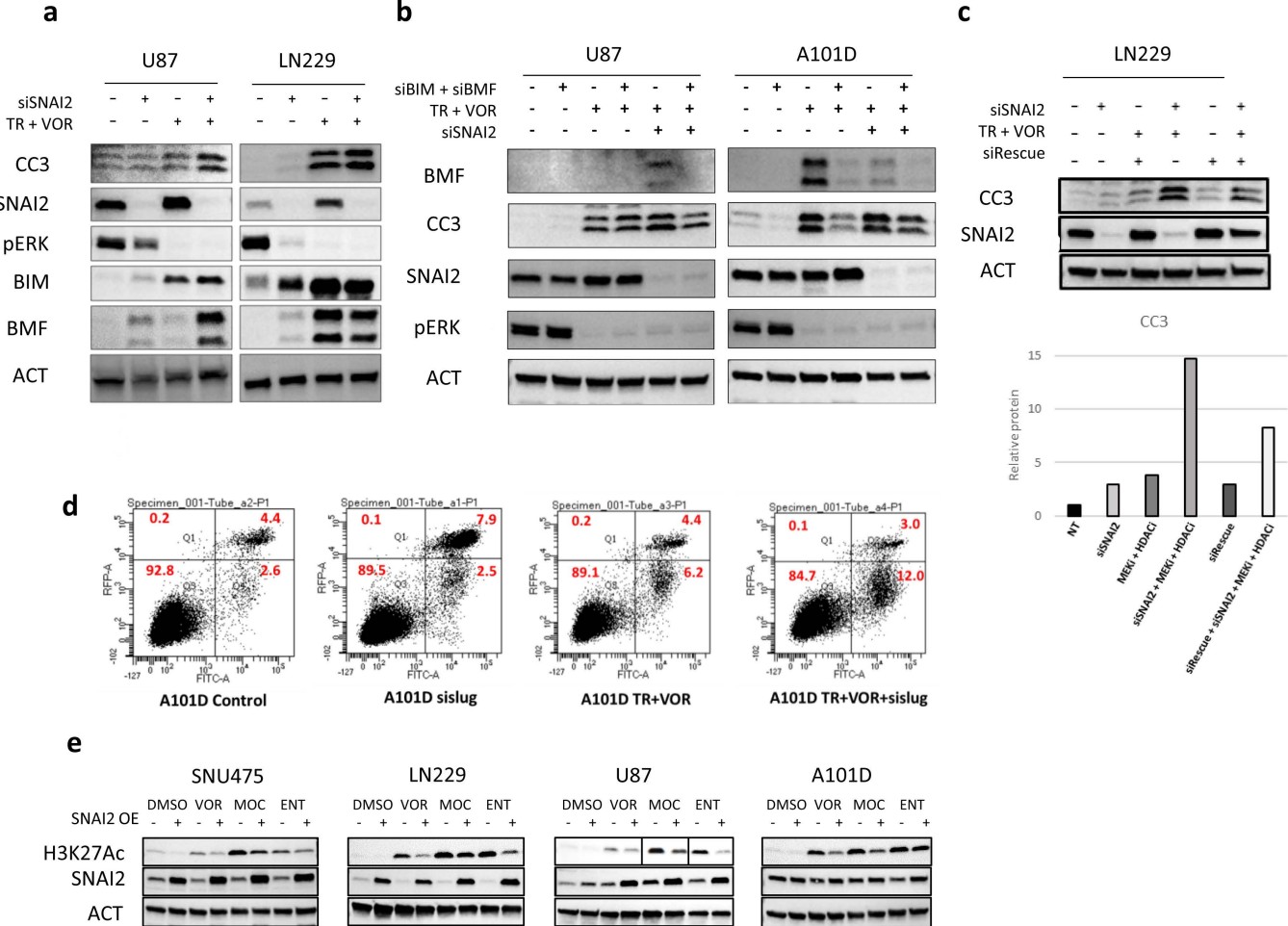

**Fig 5. SNAI2 knockdown enhances BIM/BMF dependent apoptosis in MEKi +HDACi treated TPM cells.** (a) To test the cooperativity in apoptosis induction, U87 and LN229 cells were transfected with 50 pmol SNAI2 siRNA for 72 hours followed by a 24-hour treatment with MEK1/2 and HDAC inhibitors (trametinib and vorinostat). Immunoblots were probed for markers of apoptosis induction BIM, BMF, and CC3. (b) The role of BIM and BMF in apoptosis induction upon SNAI2 KD, MEKi and HDACi was confirmed by treating the triple drug combination cells (U87 and A101D) with siRNAs against BIM and BMF. Immunoblots depict the changes in cleaved caspase-3, SNAI2, BMF and pERK. (c) SNAI2 knockdown induced acetylation and subsequent apoptosis in MEKi+HDACi treated LN229 cells was rescued using transient transfection with siSNAI2 resistant plasmid (siRes). Immunoblots depict the rescue of apoptosis (CC3). (**d**) 106 A101D cells were treated with vehicle, siSNAI2, TR + VOR or TR + VOR + siSNAI2. Cells were harvested and cell death was analyzed by flow cytometry with Annexin V-FITC and propidium iodide (PI). $10^4$ events were recorded for each sample. (e) Acetylation induced by inhibitors (vorinostat, mocetinostat and entinostat) was rescued by SNAI2 overexpression in Dox-inducible A101D and SNU475 cells (200 ng/ml doxycycline for 96 hours). Immunoblots depicting the overexpression of SNAI2 and concomitant rescue of deacetylation measured by H3K27Ac. Non-targeting siRNA pool was used as a control for each of the siRNA experiments. β-Actin was used as a loading control for immunoblots.

was confirmed by treating the triple combination cells (U87 and A101D) with siRNAs against BIM and BMF (Fig 5b). Suppression of these pro-apoptotic proteins reduced apoptosis in the treated cells, highlighting their role in this axis. To further validate the role of SNAI2 in mediating resistance to apoptosis, we performed rescue experiments using a siRNA-resistant SNAI2-expressing plasmid. Cells transfected with this resistant plasmid exhibited a marked reduction in apoptosis (CC3) compared to cells with SNAI2 knockdown alone (Fig 5c). Flow cytometry analysis of cells treated with TR + VOR and SNAI2 repression further corroborated these results, showing a higher proportion of Annexin V and propidium iodide (PI) positive cells in the triple combination treated cells, indicating increased apoptosis (Fig 5d).

HDACs comprise a large family of enzymes with overlapping substrate activity. Our previous studies indicated that TPM cancers may rely on specific HDAC enzymes to regulate cell survival [30,32]. To gain insight into the HDACs that cooperate with SNAI2 TPM cancers, we tested the ability for SNAI2 overexpression to counteract inhibitors that selectively target subsets of HDACs. These experiments indicated that total cellular levels of H3K27ac induced by various HDAC inhibitors, such as vorinostat, mocetinostat, and entinostat, were each partially rescued by overexpression of SNAI2 in SNU475, LN229, U87 and A101D cells (Fig 5e). This broad reversal of acetylation highlights the widespread role of SNAI2 in maintaining a repressive chromatin state, in part through promoting the recruitment of HDACs.

## Discussion

Although cancers with *TERT* promoter mutations occur across a seemingly disparate subset of cancer types, they are now known to be common in those that display RAS pathway expression profiles and mesenchymal traits [14]. Previous studies in selected cancer types have shown that reliance on RAS signaling and mesenchymal programs lead to specific therapeutic vulnerabilities [33,34]. We found that TPM cancers display acute sensitivity to MEK1/2 inhibition, which leads to loss of growth coupled with apoptosis caused by rescue of pro-apoptotic proteins. Similarly, suppression of anti-apoptotic BCL2 proteins (which normally sequester BIM) greatly enhanced apoptosis. We also found TPM cancers broadly respond to a combination of MEK1/2 inhibition and HDAC inhibition by undergoing elevated apoptosis. These results corroborate and extend recent reports in other mesenchymal cancers [19,35].

The basis for elevated RAS pathway activation in TPM cancers is not always obvious, as many of these tumors lack hyperactivating mutations in key RAS pathway effectors such as BRAF [4]. However, in many TPM lines, this RAS-activated phenotype may result instead from gene copy number changes or hyperactivating mutations in upstream receptors such as tyrosine kinases, or RAS gene promoter transcriptional dysregulation.

A number of studies have demonstrated that MEK1/2 inhibition does not promote DNA damage. In contrast, RAS pathway mutant melanoma cells frequently rely on RAS pathway signaling to facilitate DNA repair by driving gene expression of major DNA repair factors such as BRCA2, BRIP1/FANCJ, and XRCC5 [19]. Melanoma cells are predominantly *TERT* promoter mutant cancers and are typically RAS driven. Thus, we were surprised to find that the DNA damage we observed following low-dose TR treatment was almost exclusively due to apoptotic processes and not a direct result of inhibiting MEK1/2. One difference from the previous, we did not simultaneously inhibit BRAF with MEK1/2 [19] suggesting that BRAF may have downstream effects that are not phenocopied by MEK1/2 signaling alone.

Our study highlights the synergistic effect of combining MEK1/2 inhibition with HDAC inhibition, and we further explored the role of SNAI2 suppression in enhancing this synergy. SNAI2 is a transcription factor known for its role in epithelial-to-mesenchymal transition (EMT), which is often associated with increased invasiveness and resistance to apoptosis in cancer cells [36]. In TPM cancers, we observed that SNAI2 expression is elevated, suggesting a possible role in the aggressive phenotype of these cancers. Our results indicated for the first time that SNAI2 suppresses apoptosis by recruiting histone deacetylases (HDACs) to the promoters of key pro-apoptotic genes, such as BIM and BMF. This recruitment results in histone deacetylation, leading to a closed chromatin configuration and transcriptional repression of these genes.

The combination of MEKi and HDACi with SNAI2 knockdown led to an increase in apoptosis, indicating that SNAI2 plays a crucial role in maintaining cell survival by repressing pro-apoptotic genes. This multi-targeted approach ensures a comprehensive blockade of the survival mechanisms in TPM cancer cells and may help to prevent the emergence of resistant tumor cells following therapy, which is a common challenge with monotherapies or even dual-target therapies [37]. The benefits of this triple combination treatment over existing therapies are multifaceted. Firstly, it allows for the use of lower doses of each drug, thereby minimizing toxicity and adverse effects often associated with higher doses [38]. Secondly, it addresses the heterogeneity within tumors by targeting various cellular pathways, which may reduce the likelihood of cancer relapse. Thirdly, by attacking the cancer cells' survival mechanisms, it enhances the overall apoptotic response, leading to more effective tumor eradication.

In conclusion, our study unravels a novel role of MEK1/2, HDACs, and SNAI2 in orchestrating survival of a specific class of cancers, driving the expression of two well-known mediators of apoptosis in aggressive cancers, BIM and BMF. Our study provides a new direction for the development of therapeutic interventions against these malignancies.

## Methods

### Cell culture

*TERT* promoter mutant (TPM) cell lines used in the study were derived from Breast cancer (MDA-MB-231), hepatocellular carcinoma (SNU-475, SNU-423), melanoma (A101D (CRL-7898), UACC-257), Bladder cancer (SCaBER) and central nervous system (U87, LN229 (CRL-2611), SKNSH and DAOY (HTB-186)). Cells were maintained as an adherent culture in DMEM media (Cytiva, SH30022.02) supplemented with glutamine (Cytiva, SH30590.01), 10% fetal bovine serum (FBS) (Corning 35–010-CV) and 1% Penicillin/Streptomycin (Cytiva SV30010), sodium pyruvate (Cytiva, SH30239.01). For routine maintenance, cells were grown at 37° C and 5% $CO_2$ in a humidified atmosphere and passaged at 90% confluence by rinsing gently with 5 mL phosphate buffered saline (PBS) and incubated with 1–2 mL Trypsin-EDTA (0.25%) (Corning, 25–053-CI) at 37° C, 5% CO2 for 5 min before splitting. For maintenance of doxycycline-inducible cell lines, Tet-Free serum (Peak Serum, PS-FB3) was used for media preparation.

### Transfection and generation of stable cell lines

siRNA transfection was carried out in 6-well plates using Lipofectamine RNAiMAX transfection reagent (Thermo Fisher, 13778150). Cells were plated a day before transfection in antibiotic free media ensuring approximately 60% confluency at the time of transfection. Transfection mix was prepared as per product protocol in OPTIMEM reduced serum media (Thermo Fisher, 31985070) with 50 pmol of target siRNA (or non-targeting pool (control). SNAI2 siRNA target sequences were UCUCUCCUCUUUCCGGAUA, GCGAUGCCCAGUCUAGAAA, ACAGCGAACUGGACACACA, GAAUGUCUCUCCUGCACAA. Cells were harvested 96 hours after transfection and analyzed by immunoblotting or RT-PCR.

For generation of doxycycline inducible cell lines, SNAI2 cDNA was amplified from U87 cells and directionally inserted into a pCW backbone (Addgene, 50661) rtTA-advanced tetracycline-ON vector system using NheI and BamHI sites. The resulting plasmid was sequenced for SNAI2 to ensure error-free cloning. Lentivirus was packaged and amplified in HEK293T cells by transfection of 0.6 µg of VsVg, 3 µg of δ8.9, and 6 µg of the doxycycline-inducible SNAI2 plasmid and 0.6 µg pRev in a 100 mm$^2$ plate. Media containing lentivirus were harvested, filtered through a 0.45-micron filter, mixed with polybrene, and then added to cells. After 48 hours, lentivirally transformed cells were selected using 0.5–1 mg/ml of puromycin (Cayman Chemical Co., 13884). After puromycin selection, cells were maintained in Tet-Free media and induced with 200 ng/ml doxycycline for SNAI2 overexpression. SNAI2 siRNA ACAGCGAACUGGACACACA was used to design a rescue plasmid by mutating the sequence to ACAGCGAACU**T**GA**T**AC**G**CA.

### Cell line fingerprinting

Cell lines used in the study were authenticated by STR (short tandem repeat) analysis. Genomic DNA was extracted from the cells using the Quick-DNA Miniprep kit (Zymo Research D4068) and samples submitted to the Genomics Core facility (Heflin Center for Genomic Sciences, UAB, Alabama) for profiling of STRs (short tandem repeats). Results were confirmed upon a 100% match of at least 14 known STRs for the respective cell line.

### Immunoblot analysis

Cell lysis was performed with RIPA buffer (150 mM NaCl, 50 mM Tris, 5 mM EDTA, 1% NP-40, 0.5% Sodium deoxycholate, 0.1% SDS; pH 7.4) with protease inhibitors (PI, Thermo Fisher Scientific, A32963) using a pellet of 5 million cells

incubated on ice for 20 minutes with intermittent vortexing. The lysates were centrifuged at 13,000 x g for 20 min at 4° C. Protein concentration was determined by DC protein assay (BioRad 500–0116). Supernatant containing the solubilized proteins was resolved by sodium dodecyl sulfate (SDS) polyacrylamide gel electrophoresis (PAGE) using Bio-Rad Mini-PROTEAN TGX Stain-free pre-cast gels (4–20% gradient, 15-well, 4561096). Equal masses of protein were loaded onto the gels. The separated proteins were immunoblotted on Nitrocellulose membranes (0.2 µm BioRad, 1704158) according to the BioRad Trans-Blot Transfer using the 5-minute protocol and blocked in StartingBlock (Thermo Fisher, 37543) TBS blocking buffer for 30 min at room temperature. Membranes were incubated overnight at 4° C in primary antibody in StartingBlock. Membranes were washed three times with TBS-Tween (0.01%) with moderate agitation 3 x 5 min. The primary antibody-stained membranes were incubated with an HRP-conjugated secondary antibody (see antibody table for catalogue number), washed three times with TBS-T with moderate agitation 3 x 5 min and subsequently analyzed using enhanced chemi-luminescent substrate (ECL) (Thermo Fisher Scientific, 34578). Chemiluminescence was visualized using an iBright Imaging System (Invitrogen). Quantification of band intensity was performed with ImageJ software (NIH, http://rsbweb.nih.gov/ij/). Secondary antibody stripping was performed with Restore PLUS Immunoblot Stripping Buffer (Thermo Fisher Scientific, 46430) according to manufacturer's protocol or a low pH stripping buffer (25 mM glycine, 1% SDS; pH 2.3) for 15 minutes with moderate agitation. Briefly, the probed membranes were incubated in the stripping buffer for 20 minutes at room temperature with gentle agitation. Membranes were washed three times with PBS-T, 15 min each with moderate agitation and incubated with the next desired primary antibody. Antibodies used for immunoblots were: Cell Signaling Technologies - ERK1/2 p44/42 MAPK (Erk1/2) (#9102S), phospho-ERK1/2 (pERK; Phospho-p44/42 MAPK (Erk1/2) (Thr202/Tyr204) (#4370S), BIM (#2933), BMF (#50542), SNAI2 (#9585), Phospho-Histone H2A.X (pH2AX; (Ser139) (#9718), Cleaved Caspase-3 (CC3, #9664), Acetyl-Histone H3 (H3K9ac; Acetyl-Histone H3 (Lys9/Lys14) Antibody #9677), Acetyl-Histone H3 (H3K27ac; Acetyl-Histone H3 (Lys27) #8173), BCL2 (#4223), MCL1 (#94296), Bcl-xL (#2764), GAPDH (#5174S) and β-Actin HRP Antibody (sc-47778).

## Quantitative RT PCR analysis

Total RNA was extracted from cells using Trizol or IBI isolate (IBI Scientific, IB47602). 1 mL of TRIZOL or IBI isolate was incubated with approximately 5 million cells for 5 minutes. 200 µl of chloroform was added to each sample then mixed before incubating for 2 minutes. Samples were spun down at 12,000 x g for 15 minutes. The aqueous layer was transferred to a new tube. RNA was precipitated by adding 500 ml of isopropanol and 1 mg of glycogen (Sigma-Aldrich, 10901393001) to each sample. Samples were left at −80° C overnight. Precipitated RNA was spun down at 13,000 x g for 20 minutes at 4° C. Residual ethanol was removed, and the precipitated RNA was reconstituted in 20 µl of $H_2O$. Isolated RNA was subjected to DNase treatment (DNase1, NEB M0303L). After DNase treatment, volume was made up to 100 µL by adding 65 µL TE (pH 7) + 10 uL 3M NaOAc pH 5.2. RNA was further purified by adding 100 µl of phenol:chloroform (Thermo Fisher Scientific, BP1752I, pH 6.5) to each sample, mixing, then spinning down at 5,000 x g for 5 minutes. The aqueous layer was placed in a new tube along with 100 ml of Isoamyl-alcohol:chloroform (Thermo Fisher Scientific, X205). Samples were mixed then spun down at 5,000 x g for 5 minutes. The aqueous layer was added to a separate tube with 250 µl of ethanol. Samples were left at −80°C overnight. Precipitated RNA was spun down at 15,000 x g for 20 minutes then washed with 75% ethanol and spun down again at 15,000 x g for 5 minutes. Residual ethanol was removed, and the precipitated RNA was reconstituted in 20 µl of $H_2O$. 1 µg of resulting purified RNA was used to prepare cDNA with the help of ProtoScript II Reverse transcriptase (New England Biolabs M0368) according to manufacturer's instructions. For expression analysis by qPCR, we used SYBR Select Master Mix (Applied Biosystems) with 1 µL template cDNA and primer concentration 5 µM. Primer sequences are provided below. All PCR products were verified for specificity by agarose gel electrophoresis and DNA sequencing. PCRs were run on Bio-Rad CFX Connect RT-PCR Detection system at a pre-set melting cycle with annealing temperature specific for primer set. For quantification, Ct values were assessed using the BioRad CFX Maestro software. GAPDH RNA expression was used as an endogenous reference. Expression data was

quantified using $2^{-\Delta\Delta CT}$ method and stated as fold change in gene expression for each individual gene. Primer sequences are given below:

| Primer | Sequence |
| --- | --- |
| 18s forward | GTAACCCGTTGAACCCCATT |
| 18s reverse | CCATCCAATCGGTAGTAGCG |
| SNAI2 Forward | ACAGCAGCCAGATTCCTCAT |
| SNAI2 Reverse | CTTTTTCTTGCCCTCACTGC |
| BIM Forward | AAACCAACAAGACCCAGCAC |
| BIM Reverse | CGGTGTCTTCTGAAACGTCA |
| BIM CHIP Forward | TAG GGT ACA CTT CGG GGT GG |
| BIM CHIP Reverse | CTG GCG TGT TTA CCG GAG TA |
| BMF CHIP Forward | CTT AGC TGT TCA GGT GGT TGC |
| BMF CHIP Reverse | TTC CTC TTG CTC CAC CTG ATG |

## Growth assays

To assess growth dynamics, cells were plated at densities of 5,000 cells per well in 96-well plates. After 12 hours of incubation, cells were treated with trametinib, vorinostat, or a combination of the two, with vehicle control as a reference. Imaging was performed every 12 hours over a period of 6.5 days using a BioTek Cytation 5 automated imaging system paired with a BioSpa 8 incubator, capturing images at 10 × magnification. Cell nuclei were quantified using BioTek Gen5 software, employing a 200 µm offset for image analysis to estimate cell counts.

## Flow cytometry

Vehicle (DMSO) or drug (trametinib, vorinostat and/or siSNAI2) treated cells were harvested with mild trypsinization (2–3 minutes incubation) and tested for viability with trypan blue. Cells were washed once with ice cold PBS and then processed as per manufacturer's protocol for staining with FITC-AnnexinV and Propidium Iodide (FITC Annexin V apoptosis detection kit I, BD Biosciences, 556547) for detection of apoptosis (FITC). Propidium Iodide (PI) was used for staining necrotic cells. Flow cytometry measurements were performed on BD LSRII flow cytometer using BD FACS Suite application software. At least 10,000 cells were analysed for each sample and gating was applied to the populations to exclude cell debris and doublets. Data generated from positive gating was analysed using FlowJo software (Tree Star, USA). Flow cytometry data analysis was performed after gating on live cells in FlowJo flow cytometry analysis software. Apoptosis was quantified as percentage of AnnexinV and PI-stained cells (Q2 and Q3) obtained for each treatment and normalized to control (DMSO) samples.

## Chromatin immunoprecipitation (ChIP)

For ChIP, cells were plated in 10 cm culture plates at 60% confluency and treated with non-targeting siRNA or SNAI2-siRNA. For transfection, 2 ml transfection reagent containing 200 pmol siRNA and 20 µl RNAiMAX was added to 8 ml antibiotic-free media. The following day the media was replaced with antibiotic containing media.

At the end of transfection period of 96 hours, cells were rinsed with PBS, crosslinked with 1% Formaldehyde for 10 mins and quenched with 125 mM glycine. Cells were scraped in ice-cold PBS, centrifuged to remove PBS and the pellet incubated in -80° C for at least 1 hr. The frozen pellet was lysed in 1 ml lysis buffer (50 mM Tris-Cl pH 8, 10 mM EDTA, 0.5% SDS, 1 x protease inhibitors). Lysates were sonicated in 1.5 mL Eppendorf tube in a BioRuptor 5 x 10 min on "high" power setting, with cycles of 30 seconds on and 30 seconds off. Lysates were cleared by centrifugation at 16,000 x g for 15 minutes at 4° C and quantified with a nanodrop. 10 ug chromatin (based on Nanodrop readings for nucleic acid)

were immunoprecipitated with 2 mg of indicated antibodies and incubated overnight in a nutator at 4° C. 1 mg input samples were used for RT-PCR normalization and samples incubated with normal rabbit IgG were used as negative control. Immunoprecipitation was performed with 20 µl of protein A/G magnetic beads (Pierce, 88803) were used to capture the antibody-chromatin complex by incubation for 3 hours at 4° C on a nutator. Bead complex was sequentially washed with Low salt (20 mM Tris-Cl pH 8.0, 2 mM EDTA, 150 mM NaCl, 0.1% SDS, 1% Triton X-100), High Salt (20 mM Tris-Cl pH 8.0, 2 mM EDTA, 500 mM NaCl, 0.1% SDS, 1% Triton X-100), LiCl (10 mM Tris-Cl pH8.0, 1 mM EDTA, 250 mM LiCl, 1% deoxycholate, 1% NP40) and TE (10 mM Tris pH 8, 2 mM EDTA) Buffers and eluted with 100 mM NaHCO3 and 1% SDS. Crosslinks were reversed by the addition of 5 M NaCl overnight at 65° C. Protein and RNA in the sample were digested by adding 7 µL of 1 M Tris pH 6.5, 3 µL of 500 mM EDTA, 3 µL of proteinase K (20 mg/mL), 0.5 µL of concentrated RNase A (10 mg/mL, Thermo Fisher EN0531) and incubating at 37°C for 60 minutes. DNA was purified using phenol-chloroform (Thermo Fisher Scientific, BP1752I, pH 8) and chloroform isoamyl alcohol extraction followed by the addition of 0.1 volumes 3 M NaOAc, 0.5 µg glycogen and 1 mL of ethanol precipitation, washed 1 x with 70% ethanol. The precipitated DNA was reconstituted in TE buffer and used for qPCR (primers for targets given in Supplementary Material). For qPCR, 1 µL was used in a 10 µL reaction. ChIP samples were normalized to input samples.

## Supporting information

**S1 File.** Fig S1. ERK activation in TPM Cancer Cells is Inhibited by Low Doses of MEK1/2 inhibition (MEKi). Fig S2. BIM protein is below average in most TPM cancer types. Fig. S3 Combining MEKi and HDACi induces apoptosis in TPM cells. Table S1. Cell lines used in this study. Example gating strategy used for Flow cytometry.
(PDF)

## Acknowledgments

C. Ryan Miller and Erin Smithberger, O'Neal Comprehensive Cancer Center, University of Alabama at Birmingham; James Costello, University of Colorado, Anschutz.

## Author contributions

**Conceptualization:** Amol Tandon, Josh Lewis Stern.

**Investigation:** Amol Tandon, Josh Lewis Stern.

**Methodology:** Amol Tandon.

**Project administration:** Josh Lewis Stern.

**Supervision:** Josh Lewis Stern.

**Writing – original draft:** Amol Tandon.

**Writing – review & editing:** Amol Tandon, Josh Lewis Stern.

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
