## [Decision Letter · Decision Letter 0]

PONE-D-24-48344SNAI2 cooperates with MEK1/2 and HDACs to suppress BIM- and BMF-dependent apoptosis in TERT promoter mutant cancersPLOS ONE

Dear Dr. Stern,

Thank you for submitting your manuscript to PLOS ONE. After careful consideration, we feel that it has merit but does not fully meet PLOS ONE’s publication criteria as it currently stands. Therefore, we invite you to submit a revised version of the manuscript that addresses the points raised during the review process.

We look forward to receiving your revised manuscript.

Kind regards,

Kai Huang

Academic Editor

PLOS ONE

Journal Requirements:

3. Please update your submission to use the PLOS LaTeX template. The template and more information on our requirements for LaTeX submissions can be found at http://journals.plos.org/plosone/s/latex .

5. We note that your Data Availability Statement is currently as follows: [All relevant data are within the manuscript and its Supporting Information files.)

6. PLOS requires an ORCID iD for the corresponding author in Editorial Manager on papers submitted after December 6th, 2016. Please ensure that you have an ORCID iD and that it is validated in Editorial Manager. To do this, go to ‘Update my Information’ (in the upper left-hand corner of the main menu), and click on the Fetch/Validate link next to the ORCID field. This will take you to the ORCID site and allow you to create a new iD or authenticate a pre-existing iD in Editorial Manager.

7. We note that you have included the phrase “data not shown” in your manuscript. Unfortunately, this does not meet our data sharing requirements. PLOS does not permit references to inaccessible data. We require that authors provide all relevant data within the paper, Supporting Information files, or in an acceptable, public repository. Please add a citation to support this phrase or upload the data that corresponds with these findings to a stable repository (such as Figshare or Dryad) and provide and URLs, DOIs, or accession numbers that may be used to access these data. Or, if the data are not a core part of the research being presented in your study, we ask that you remove the phrase that refers to these data.

8. Please include captions for your Supporting Information files at the end of your manuscript, and update any in-text citations to match accordingly. Please see our Supporting Information guidelines for more information: http://journals.plos.org/plosone/s/supporting-information

Reviewers' comments:

Reviewer's Responses to Questions

**Comments to the Author**

1. Is the manuscript technically sound, and do the data support the conclusions?

Reviewer #1: Yes

Reviewer #2: Yes

2. Has the statistical analysis been performed appropriately and rigorously? 

Reviewer #1: Yes

Reviewer #2: Yes

3. Have the authors made all data underlying the findings in their manuscript fully available?

Reviewer #1: Yes

Reviewer #2: Yes

4. Is the manuscript presented in an intelligible fashion and written in standard English?

Reviewer #1: Yes

Reviewer #2: Yes

5. Review Comments to the Author

Reviewer #1: In this manuscript, the authors first demonstrated that the combination of MEK inhibitors (MEKi) and histone deacetylase inhibitors (HDACi) can enhance apoptosis in multiple TPM cancer cell lines. They further confirmed that the induced apoptosis is dependent on BIM and BMF, and that this effect can be augmented by SNAI2 knockdown. Overall, this is a well-structured and well-executed study that can provide potentially valuable insights in the field.

However, there are several points requiring further confirmation and clarification of their results. Below, I outline both major and minor concerns that should be addressed to strengthen the manuscript.

Major concerns:

1. The authors demonstrated that pErk is activated in multiple TPM cancer cell lines. However, how does this compare in non-TPM cancer cells and/or healthy cells? The same question applies to the EMT markers examined by the authors. Additionally, does the combination of MEKi and HDACi have similar effects on apoptosis in non-TPM cancer cells? If so, is it still BIM/BMF/SNAI2-dependent?

2. In addition to BIM, the authors should also comment on the expression levels of other apoptosis-related proteins, such as BCL-2, BCL-XL, BAX, and BAK, in TPM cancer cells compared to non-TPM cancer cells and/or healthy cells. Furthermore, the authors should explain why they chose to focus on BIM rather than other apoptosis-related proteins.

3. The authors primarily used cleaved caspase-3 by western blotting as a marker for apoptosis. However, it is recommended that they confirm some of their key results using more direct methods to assess apoptosis, such as flow cytometry with Annexin V-PI staining. This suggestion applies to, but is not limited to, the examination of the effects of BIM and BMF on MEKi+HDACi.

4. The legend for Figure 2c states that cell counts were performed using an automated cell counter (BioSpa Cytation-5). However, the main text mentions that cell growth was measured using CellTiter-Glo. Additionally, the inhibitory effects of each individual drug alone are already quite strong, making it difficult to discern whether there are any additional benefits when the two drugs are combined in Figure 2c.

Minor concerns:

1. All western blot results should be accompanied by at least one molecular weight marker.

2. Some of the figures are labeled as "slug," while others are labeled as "SNAI2." The authors should use consistent labeling/annotation throughout the manuscript.

3. In Figure 5e, SNAI2 does not appear to be overexpressed in U87 and A101D cells.

4. The authors should provide the sources of the cell lines used in the manuscript.

5. The authors should provide the sources of the lentiviral packaging plasmids used in the manuscript.

6. The authors should provide the sources of all antibodies used in the manuscript.

7. The authors should provide the sources and sequences of all primers used in the manuscript.

8. The authors should provide examples of the gating strategies used for their flow cytometry analysis in supplementary.

Reviewer #2: Major comments

1. Line 91-94. The authors stated, 'We first validated RAS pathway activation in a panel of TPM cancer cell lines by determining the level of phosphorylated ERK1 and ERK2 (pERK). We tested melanoma (A101D, UACC257), hepatocellular carcinoma (SNU475, SNU423), breast cancer (MM231), medulloblastoma (DAOY), and neuroblastoma (SKNSH) with mutations in the TERT promoter.' It appears that the authors intended to demonstrate that TERT promoter mutations (TPM) activate the RAS pathway by measuring pERK levels (Figure S1a). However, it may not be appropriate to include melanoma (A101D, UACC257), breast cancer (MM231), and neuroblastoma (SKNSH) cell lines, as these lines have not only TPM but also mutations in the RAS/RAF/ERK pathway, as indicated in Table S1. Specifically, A101D (BRAF Val600Glu), UACC257 (BRAF Val600Glu), MM231 (assumed to be MDA-MB-231, KRAS Gly13Asp), and SKNSH (NRAS Gln61Lys) possess mutations that could be known to activate the RAS pathway (reflected by pERK signaling), making it difficult to attribute the observed pathway activation solely to the TERT promoter mutation.

2. To follow up on major comment 1, it would be very helpful to provide solid evidence by correcting the TERT promoter mutation using CRISPR techniques such as base editing or knock-in in these cell lines. This approach would provide direct evidence to support the study's conclusions. It is crucial to clarify which mutations play a major role in tumor progression. If the authors intend to investigate the function of the TERT promoter mutation, it may be advisable to avoid using RAS/RAF/ERK-mutant cell lines. Alternatively, acutely correcting the TERT mutation could help determine if cell viability decreases, this will reveal the mutation's role in oncogenesis.

Minor comments

1. Please provide the detailed information of MM231 in the methods part, if “MM231” is an abbreviation for “MDA-MB-231”

2. Please clarify the pERK phosphorylation site in the manuscript and label it in Figure S1a. Additionally, ensure consistent formatting of 'ACT' and 'Actin' in Figure S1.

3. Please clarify Figure 3b and 3c—why some of the column plots do not have error bars. Additionally, could the authors clarify how the 'relative CC3' value is defined?

6. PLOS authors have the option to publish the peer review history of their article (what does this mean? ). If published, this will include your full peer review and any attached files.

**Do you want your identity to be public for this peer review?** For information about this choice, including consent withdrawal, please see our Privacy Policy .

Reviewer #1: No

Reviewer #2: No

---

## [Author Response · Author response to Decision Letter 1]

17 Jan 2025

We would like to thank the reviewers for the thoughtful reading of our manuscript. Below we have addressed each of the points raised.

Reviewer #1:

Major Concerns:

1. The authors demonstrated that pErk is activated in multiple TPM cancer cell lines. However, how does this compare in non-TPM cancer cells and/or healthy cells? The same question applies to the EMT markers examined by the authors. Additionally, does the combination of MEKi and HDACi have similar effects on apoptosis in non-TPM cancer cells? If so, is it still BIM/BMF/SNAI2-dependent?:

o We acknowledge the importance of understanding whether the mechanisms we observed are specific to TPM cancer cells. In our study, we have focused on TPM cancer cell lines, given their clinical relevance and the unique therapeutic vulnerabilities they exhibit. We previously established the biological relevance of this group of cancers (Stern et al 2020, Molecular Cancer Research). That study provided the foundation for the focus of our current work on TPM cancers specifically. While the inclusion of non-TPM or healthy cell lines would provide additional context, such analyses go beyond the scope of this study. We agree with the reviewer that our current findings establish a robust framework for exploring such comparisons in future work.

2. In addition to BIM, the authors should also comment on the expression levels of other apoptosis-related proteins, such as BCL-2, BCL-XL, BAX, and BAK, in TPM cancer cells compared to non-TPM cancer cells and/or healthy cells. Furthermore, the authors should explain why they chose to focus on BIM rather than other apoptosis-related proteins:

o BIM is strongly suppressed in TPM cancers and other tumor types with activated pERK. While we centered our study on BIM due to its established role in intrinsic apoptosis and its regulation by MEK1/2, we agree that assessing other apoptosis-related proteins is valuable. Notably, we observed no significant changes in BCL-2 or BCL-XL levels under our experimental conditions (Supplementary Fig. S3b). This supports our conclusion that BIM plays a major role in the apoptotic response induced by inhibitors of MEK1/2 and HDACs in TPM cancers.

3. The authors primarily used cleaved caspase-3 by western blotting as a marker for apoptosis. However, it is recommended that they confirm some of their key results using more direct methods to assess apoptosis, such as flow cytometry with Annexin V-PI staining. This suggestion applies to, but is not limited to, the examination of the effects of BIM and BMF on MEKi+HDACi.:

o We appreciate the reviewer’s suggestion to use Annexin V-PI flow cytometry, which we indeed performed and presented in the manuscript (Fig. 5d). These data demonstrate an increase in apoptotic cells following the combination treatment and confirm the cleaved caspase-3 results. The inclusion of multiple methods strengthens the rigor of our conclusions.

4. The legend for Figure 2c states that cell counts were performed using an automated cell counter (BioSpa Cytation-5). However, the main text mentions that cell growth was measured using CellTiter-Glo. Additionally, the inhibitory effects of each individual drug alone are already quite strong, making it difficult to discern whether there are any additional benefits when the two drugs are combined in Figure 2c:

o We thank the reviewer for bringing this to our attention. We used automated cell counting to assess growth. We have revised the figure legend and text to clarify this distinction. Additionally, we recognize that individual drugs show significant inhibitory effects on growth (Fig. 2c) and that these are distinct from the combinatorial effects we observe on cell death, and on BIM expression.

Minor Concerns:

1. All western blot results should be accompanied by at least one molecular weight marker.

2. Some of the figures are labeled as "slug," while others are labeled as "SNAI2." The authors should use consistent labeling/annotation throughout the manuscript.

3. In Figure 5e, SNAI2 does not appear to be overexpressed in U87 and A101D cells.

4. The authors should provide the sources of the cell lines used in the manuscript.

5. The authors should provide the sources of the lentiviral packaging plasmids used in the manuscript.

6. The authors should provide the sources of all antibodies used in the manuscript.

7. The authors should provide the sources and sequences of all primers used in the manuscript.

8. The authors should provide examples of the gating strategies used for their flow cytometry analysis in supplementary.

1. We have provided uncropped images of all blots containing the molecular weight markers.

2. We have ensured consistent use of “SNAI2” instead of “slug.”

3. For Figure 5e, the variations in SNAI2 expression reflects differences in cell line-specific regulation and technical variability (exposure of western blot membrane). This has been addressed in the text.

4. 4-7. Sources for cell lines, plasmids, antibodies, and primers have been added to the Methods section.

5. Gating strategies for flow cytometry are included in the Supplementary Material.

Reviewer #2:

Major Comments:

1. Line 91-94. The authors stated, 'We first validated RAS pathway activation in a panel of TPM cancer cell lines by determining the level of phosphorylated ERK1 and ERK2 (pERK). We tested melanoma (A101D, UACC257), hepatocellular carcinoma (SNU475, SNU423), breast cancer (MM231), medulloblastoma (DAOY), and neuroblastoma (SKNSH) with mutations in the TERT promoter.' It appears that the authors intended to demonstrate that TERT promoter mutations (TPM) activate the RAS pathway by measuring pERK levels (Figure S1a). However, it may not be appropriate to include melanoma (A101D, UACC257), breast cancer (MM231), and neuroblastoma (SKNSH) cell lines, as these lines have not only TPM but also mutations in the RAS/RAF/ERK pathway, as indicated in Table S1. Specifically, A101D (BRAF Val600Glu), UACC257 (BRAF Val600Glu), MM231 (assumed to be MDA-MB-231, KRAS Gly13Asp), and SKNSH (NRAS Gln61Lys) possess mutations that could be known to activate the RAS pathway (reflected by pERK signaling), making it difficult to attribute the observed pathway activation solely to the TERT promoter mutation:

o We thank the reviewer for their careful assessment of our manuscript and for highlighting the potential confounding factors related to the inclusion of certain cell lines with RAS/RAF pathway mutations. Our goal was not to assert a causal relationship between TERT promoter mutations (TPMs) and RAS pathway activation. Rather, we aimed to demonstrate with direct molecular evidence the association between TPMs and elevated RAS pathway signaling as previously described through bioinformatic assessment (Stern et al 2020, Molecular Cancer Research) and show here molecular evidence of activated ERK in TPM cancers. As noted in our manuscript (Lines 91–94), all cell lines examined possess TERT promoter mutations, and the high levels of pERK observed across this diverse panel, including lines with and without RAS/RAF mutations, suggest a broader association of TPMs with RAS pathway activation. We acknowledge that the inclusion of cell lines with RAS/RAF pathway mutations (e.g., A101D, UACC257, MM231, and SKNSH) complicates causal interpretations. However, our study does not seek to isolate TPMs as the sole driver of RAS pathway activation but rather to highlight their frequent co-occurrence with this signaling phenotype.

o

To ensure clarity, we have revised the text as:

"We first validated RAS pathway activation in a panel of TPM cancer cell lines by determining the level of phosphorylated ERK1 and ERK2 (pERK), aiming to document the frequent association of TPMs with elevated RAS signaling. This panel included melanoma (A101D, UACC257), hepatocellular carcinoma (SNU475, SNU423), breast cancer (MDA-MB-231), medulloblastoma (DAOY), and neuroblastoma (SKNSH), which represent diverse cancer types harboring TPMs. It should be noted that some of these lines (A101D, UACC257, MM231, and SKNSH) also carry activating mutations in the RAS/RAF/ERK pathway, as detailed in Table S1."

2. To follow up on major comment 1, it would be very helpful to provide solid evidence by correcting the TERT promoter mutation using CRISPR techniques such as base editing or knock-in in these cell lines. This approach would provide direct evidence to support the study's conclusions. It is crucial to clarify which mutations play a major role in tumor progression. If the authors intend to investigate the function of the TERT promoter mutation, it may be advisable to avoid using RAS/RAF/ERK-mutant cell lines. Alternatively, acutely correcting the TERT mutation could help determine if cell viability decreases, this will reveal the mutation's role in oncogenesis.:

o We thank the reviewer for their thoughtful suggestion and for highlighting the potential utility of CRISPR-based approaches to dissect the role of the TERT promoter mutation (TPM). Importantly, the role of TPMs in driving telomerase activity and enabling oncogenesis is well-documented (Horn et al., 2013; Vinagre et al., 2013, Stern et al., 2015, Stern et al., 2017, Stern et al., 2020). Instead, our study focuses on characterizing TPM cancers as a group with shared biological features, including elevated RAS pathway activity, and on exploring their vulnerabilities to combined MEK1/2 inhibition (MEKi), HDAC inhibition (HDACi), and SNAI2 suppression. In our study, TPM serves as a marker for a subset of cancers characterized by a RAS-activated phenotype, which we identify as being particularly sensitive to MEKi/HDACi/SNAI2-targeted therapies.

Minor Comments:

1. Please provide the detailed information of MM231 in the methods part, if “MM231” is an abbreviation for “MDA-MB-231”

2. Please clarify the pERK phosphorylation site in the manuscript and label it in Figure S1a. Additionally, ensure consistent formatting of 'ACT' and 'Actin' in Figure S1.

3. Please clarify Figure 3b and 3c—why some of the column plots do not have error bars. Additionally, could the authors clarify how the 'relative CC3' value is defined?

1. "MM231" has been clarified as "MDA-MB-231."

2. pERK phosphorylation site (Thr202/Tyr204) is stated in the text and figure legends.

3. Missing error bars have been added for clarity. The “relative CC3” levels were calculated after quantification of CC3 western blots (three biological replicates) and normalizing to the respective Actin signals. This has been added in the respective figure legend.

---

## [Decision Letter · Decision Letter 1]

PONE-D-24-48344R1SNAI2 cooperates with MEK1/2 and HDACs to suppress BIM- and BMF-dependent apoptosis in TERT promoter mutant cancersPLOS ONE

Dear Dr. STERN,

Thank you for submitting your manuscript to PLOS ONE. After careful consideration, we feel that it has merit but does not fully meet PLOS ONE’s publication criteria as it currently stands. Therefore, we invite you to submit a revised version of the manuscript that addresses the points raised during the review process.

We look forward to receiving your revised manuscript.

Kind regards,

Kai Huang

Academic Editor

PLOS ONE

Journal Requirements:

Reviewers' comments:

Reviewer's Responses to Questions

**Comments to the Author**

1. If the authors have adequately addressed your comments raised in a previous round of review and you feel that this manuscript is now acceptable for publication, you may indicate that here to bypass the “Comments to the Author” section, enter your conflict of interest statement in the “Confidential to Editor” section, and submit your "Accept" recommendation.

Reviewer #1: (No Response)

Reviewer #2: All comments have been addressed

2. Is the manuscript technically sound, and do the data support the conclusions?

Reviewer #1: Yes

Reviewer #2: Yes

3. Has the statistical analysis been performed appropriately and rigorously? 

Reviewer #1: Yes

Reviewer #2: Yes

4. Have the authors made all data underlying the findings in their manuscript fully available?

Reviewer #1: Yes

Reviewer #2: Yes

5. Is the manuscript presented in an intelligible fashion and written in standard English?

Reviewer #1: Yes

Reviewer #2: Yes

6. Review Comments to the Author

Reviewer #1: I appreciate the authors' efforts in addressing my previous concerns. Overall, I find the revised manuscript to be satisfactory. However, I would like to raise a minor point regarding Figure 2C. It would be helpful to include additional zoomed-in figures at lower cell viabilities to better illustrate the combinational effects of TR+VOR on cell viability and cell growth, as these effects are not easily discernible in the current figures.

Reviewer #2: Thank you for your revision. I’m satisfied with your revision. No more comments. I look forward to seeing your upcoming research.

7. PLOS authors have the option to publish the peer review history of their article (what does this mean? ). If published, this will include your full peer review and any attached files.

**Do you want your identity to be public for this peer review?** For information about this choice, including consent withdrawal, please see our Privacy Policy .

Reviewer #1: No

Reviewer #2: No

---

## [Author Response · Author response to Decision Letter 2]

26 Mar 2025

We thank the the reviewers and have modified Fig 2c to have a log axis.

---

## [Editor Report · Decision Letter 2]

SNAI2 cooperates with MEK1/2 and HDACs to suppress BIM- and BMF-dependent apoptosis in TERT promoter mutant cancers

PONE-D-24-48344R2

Dear Dr. Tandon,

We’re pleased to inform you that your manuscript has been judged scientifically suitable for publication and will be formally accepted for publication once it meets all outstanding technical requirements.

Kind regards,

Kai Huang

Academic Editor

PLOS ONE
---

## [Editor Report · Acceptance letter]

PONE-D-24-48344R2

PLOS ONE

Dear Dr. Tandon,

I'm pleased to inform you that your manuscript has been deemed suitable for publication in PLOS ONE. Congratulations! Your manuscript is now being handed over to our production team.

Kind regards,

on behalf of

Dr. Kai Huang

Academic Editor

PLOS ONE